# Ionizing Radiation Induces Disc Annulus Fibrosus Senescence and Matrix Catabolism via MMP-Mediated Pathways

**DOI:** 10.3390/ijms23074014

**Published:** 2022-04-05

**Authors:** Jiongbiao Zhong, Joseph Chen, Anthony A. Oyekan, Michael W. Epperly, Joel S. Greenberger, Joon Y. Lee, Gwendolyn A. Sowa, Nam V. Vo

**Affiliations:** 1Ferguson Laboratory for Spine Research, University of Pittsburgh, Pittsburgh, PA 15213, USA; zhongjiongbiao@126.com (J.Z.); chen.joseph@medstudent.pitt.edu (J.C.); oyekanaa@upmc.edu (A.A.O.); leejy3@upmc.edu (J.Y.L.); sowag@upmc.edu (G.A.S.); 2Department of Orthopaedic Surgery, University of Pittsburgh, Pittsburgh, PA 15213, USA; 3Pittsburgh Ortho Spine Research (POSR) Group, University of Pittsburgh, Pittsburgh, PA 15213, USA; 4Department of Radiation Oncology, University of Pittsburgh, Pittsburgh, PA 15213, USA; eppemw@upmc.edu (M.W.E.); greenbergerjs@upmc.edu (J.S.G.); 5Department of Physical Medicine and Rehabilitation, University of Pittsburgh, Pittsburgh, PA 15213, USA

**Keywords:** aging, genotoxic stress, intervertebral disc degeneration, cellular senescence, ionization radiation, DNA damage

## Abstract

Previous research has identified an association between external radiation and disc degeneration, but the mechanism was poorly understood. This study explores the effects of ionizing radiation (IR) on inducing cellular senescence of annulus fibrosus (AF) in cell culture and in an in vivo mouse model. Exposure of AF cell culture to 10–15 Gy IR for 5 min followed by 5 days of culture incubation resulted in almost complete senescence induction as evidenced by SA-βgal positive staining of cells and elevated mRNA expression of the p16 and p21 senescent markers. IR-induced senescent AF cells exhibited increased matrix catabolism, including elevated matrix metalloproteinase (MMP)-1 and -3 protein expression and aggrecanolysis. Analogous results were seen with whole body IR-exposed mice, demonstrating that genotoxic stress also drives disc cellular senescence and matrix catabolism in vivo. These results have important clinical implications in the potential adverse effects of ionizing radiation on spinal health.

## 1. Introduction

Intervertebral disc degeneration (IDD) is a major contributor to low back pain (LBP) [1,2,3,4]. A population-based study has suggested that individuals with IDD are at three times greater risk for developing LBP [5,6]. As life expectancy increases with medical advancements, age-related degenerative disorders emerge as significant global health burdens [7,8,9,10]. Aging is a driving force behind IDD [7,11,12]. Aging drives accumulation of molecular damage leading to dysregulation of disc cellular activities and progressive loss of proteoglycan (PG), an important disc structural matrix constituent [12,13]. Age-related loss of disc PG is a result of matrix homeostatic imbalance from increased matrix catabolic activities and decreased anabolic activities due to time-dependent accumulation of mechanical tissue injuries and molecular damage [13]. Aggrecan, the most abundant and important PG in the intervertebral disc (IVD) in maintaining IVD hydration and height, undergoes multiple degenerative changes with age [14]. These modifications include proteolytic destruction of the aggrecan core protein and the glycosaminoglycan side chains, leading to loss of IVD tissue hydration and impaired mechanical function in bearing load [12,15,16,17]. The major enzymes involved in aggrecanolysis include matrix metalloproteinases (MMPs) and aggrecanases such as a disintegrin and metalloproteinase with thrombospondin motifs (ADAMTS) [18,19,20]. As our aging population grows rapidly, it is imperative that we understand the mechanisms of age-dependent IDD leading to LBP to develop effective therapeutic interventions.

Although age-dependent accumulation of macromolecular damage includes damage to cellular protein and lipid structures, DNA damage is particularly harmful and has been established as a significant cause of aging [21]. A growing body of research reveals that persistent DNA damage, driven by both endogenous [22,23,24,25] and environmental [26,27,28,29,30,31] sources of genotoxic stress, triggers signaling cascades that can drive cells into apoptosis or senescence to avoid replicating a damaged genome. Senescent cells undergo irreversible growth arrest and acquire a senescence-associated secretory phenotype (SASP) that secrets pro-inflammatory cytokines, chemokines, and tissue-damaging proteases that negatively impact matrix homeostasis, stem and progenitor cell function, hemostatic factors, and growth factors. Hence loss of cellular function from DNA damage-induced apoptosis and cellular senescence drives age-related disorders.

Growing research also demonstrates that DNA damage drives disc-specific aging principally through cellular senescence [7,13,32,33,34,35]. This is evidenced by animal models harboring increased genotoxic damage, which leads to accelerated disc cellular senescence and age-related IDD, including pronounced loss of disc PG. These models include *Ercc**1^−/∆^* mice deficient in a critical DNA repair enzyme (ERCC1) and mice exposed to genotoxic stress—such as tobacco smoke or the cancer therapeutic agent mechlorethamine (MEC) that induces DNA damage [36,37]. Indeed, DNA-induced senescent disc cells acquire SASP and exhibit pronounced matrix homeostatic imbalance from enhanced catabolic activities, including elevated aggrecanolysis [13]. These findings indicate that nuclear genome integrity is vital for disc tissue health.

Radiotherapy by ionizing radiation (IR) is a major interventional tool in cancer management [38,39,40]. Despite the growing popularity and number of indications for radiotherapy, there is still much to be known about the effects of IR as it pertains to age-related IDD. High-dose IR exposure is known to alter the architectural and structural integrity of the vertebral body [33,41,42]. However, scarce literature pertaining to the relationship between IR and progression towards IDD exists. Previous investigations primarily focused on the adverse effects of DNA damage on nucleus pulposus (NP) cells related to their senescence and PG loss but neglected the annulus fibrosus (AF) cells of the intervertebral disc (IVD). The present study sought to explore the effects of genotoxic stress, in the form of IR, on matrix homeostasis and cellular senescence using AF cell culture and mouse models.

## 2. Results

### 2.1. Induction of AF Cellular Senescence in Cell Culture by IR in a Dose-Dependent Manner

Rat AF cell cultures exposed to increasing doses of IR (0–15 Gy) exhibited several key markers of cellular senescence. AF cells administered 10 or 15 Gy displayed morphological changes 5 days after IR treatment (Figure 1A). The Cell Counting Kit 8 (CCK8) assay indicated that cell proliferation in AF cell culture was greatly reduced starting at day 5 post-IR treatment in AF cell cultures receiving 10 and 15 Gy dose (0.84 and 0.43 OD, respectively) compared to control cells without IR treatment at that timepoint (1.83 OD) (Figure 1B). AF cells treated with 10 Gy and 15 Gy were stained positive (88% and 77%, respectively) for senescence-associated β-galactosidase (SA-βgal) activity, a key marker of cellular senescence (Figure 1C). Gene transcripts for other senescence markers, the cell cycle regulators p16^INK4a^ and p21^CIP1^, increased with IR treatment in a dose-dependent relationship with highest expression at 15 Gy dosage. These RT-PCR findings were confirmed using immunofluorescence (IF) staining of p16^INK4a^ and p21^CIP1^ (Figure 1D). Based on these biochemical assays, 15 Gy was chosen as the optimal disc senescence-inducing IR dose for our subsequent experiments.

Cell culture of human AF cells exposed to IR also underwent cellular senescence. Initial imaging of these cells 5 days post-IR treatment showed a more disorganized and disarrayed culture phenotype in IR group cells than in control group cells (Appendix A
Figure A1A). Cellular senescence arose from IR-induced genotoxic stress as evident by an increase in SA-βgal staining, with 76% positive cells from IR-treated hAF cells compared to 18% positive cells from nontreated control (Appendix A
Figure A1B). These data indicate that after 5 days of exposure most AF cells in cultures exposed to a single dose of 10–15Gy IR acquire senescent cells.

### 2.2. IR-Induced Senescent AF Cells Exhibited Increased Matrix Catabolism

To investigate whether IR-induced senescent AF cells displayed matrix homeostatic imbalance, rat AF cell cultures were treated with 15 Gy IR and incubated for 5 days to establish senescence. To determine PG breakdown, we performed immunoblot analysis using antibodies against G1 of aggrecan to detect proteolytic cleavage of the IGD of aggrecan by MMP and ADAMTS, the primary PG constituent responsible for the osmotic turgidity of the disc (Figure 2A). Cleavage within the IGD is considered most pathological because it leads to loss of the entire GAG-containing region that is vital for disc function [43]. Western blot analysis revealed increased aggrecan fragmentation in the rat AF cells receiving 15Gy of IR relative to control cells (Figure 2A). IR-induced aggrecan fragmentation was predominantly observed in the MMP-mediated fragmentation compared to ADAMTS-mediated fragmentation. Matrix catabolism was further confirmed with IF staining and qRT-PCR quantification for expression of MMP-1 and -3, the two major MMPs involved in IDD [35]. There was significantly increased MMP-1 and -3 protein expression as assessed by IF (Figure 2B) and mRNA expression as measured by qRT-PCR (Figure 2C) in IR-treated rat AF cells compared to untreated control AF cells. Similarly, hAF cells exposed to IR also exhibited significantly more MMP-mediated (*p* = 0.002) and ADAMTS-mediated (*p* < 0.001) breakdown of aggrecan matrix in IR-exposed hAF cells as compared to controls (Appendix A
Figure A1C).

### 2.3. Effects of IR Treatment on Matrix Anabolism in AF Cells

AF cells exposed to 15 Gy IR showed increased matrix structural collagen-1, collagen-2, and aggrecan mRNA expression compared to untreated cells (Figure 3A). However, IF results revealed increased collagen-2 but unchanged collagen-1 and aggrecan with IR treatment, indicating the disconnect between transcription and translation of collagen-1 and aggrecan in AF cells under these conditions. The total glycosaminoglycan (GAG) content of IR-treated rat AF cells (1.8 μg GAG/ng DNA) was considerably lower than the GAG content of control AF cells (2.7 μg GAG/ng DNA). It is possible that lower GAG reflected an increase in aggrecan degradation and loss in IR-exposed AF cells.

### 2.4. Whole Body IR Treatment Resulted in Increased Intervertebral Disc Cell Senescence in Mice

To evaluate the effect of IR exposure on IVD in vivo, 5-month-old C57Bl6 mice were exposed to whole-body IR at 3 Gy daily for one week. The mice were sacrificed one year later to evaluate the effects of IR on their IVD senescence and matrix homeostasis. Increased mRNA levels of the senescence marker p21^CIP1^ were identified in IVDs of IR-exposed mice compared to untreated controls (Figure 4). High mobility group box protein 1 (HMGB1) localizes to the nuclei of normal healthy cells but is secreted into the cytoplasm upon the development of cellular senescence. Hence, loss of nuclear HMGB1 expression is a marker of cellular senescence. IR-treated mice displayed decreased disc tissue nuclear expression of HMGB1 (7.2%) compared to untreated mice (23%), suggesting that IR exposure induces disc cellular senescence in vivo (Figure 4).

### 2.5. IR-Treated Mice Exhibited Perturbed Disc Proteoglycan Homeostasis

Disc tissue from IR-treated mice exhibited lower aggrecan gene expression compared to control mouse discs (*p* < 0.001) (Figure 5), suggesting decreased matrix anabolism. Moreover, discs from IR-treated mice showed elevated ADAMTS- and MMP-mediated aggrecanolysis in similar proportions. ADAMTS-generated aggrecan fragment was 1.5 times greater in IR-treated mouse disc tissue than in control mice (*p* = 0.032), while the MMP-generated aggrecan fragment was 1.7 times greater in IR-treated mouse disc tissue compared to control mice (*p* = 0.001) (Figure 6).

## 3. Discussion

Our findings in this study uncovered important mechanistic insights into how IR-induced genotoxic stress promotes IDD. IR exposure resulted in disc cellular senescence and increased matrix catabolism. In an in vitro rat AF cell culture model, IR treatment led to dose-dependent increases in cellular senescence phenotype and MMP-mediated aggrecanolysis. These findings were further confirmed in an in vivo murine model, as IR whole- body exposure led to increased disc cellular senescence and aggrecanolysis that was both ADAMTS- and MMP-mediated.

Our results are consistent with prior evaluations of genotoxin-induced DNA damage resulting in IDD. One notable example is tobacco smoke–induced covalent DNA modification [44]. A previous study by Wang D et al. reported that mice treated with tobacco smoke showed increased MMP- and ADAMTS-mediated aggrecanolysis, increased expression of the senescent marker p16, and reduced PG synthesis in their IVDs [37]. Further, tobacco smoke in humans has been suggested by Battie et al. as an underlying contributor to IDD [45]. Together, these findings implicate DNA damage as a mechanism towards IDD.

Ionizing radiation (IR) causes direct, physical DNA damage by ionizing or breaking DNA molecules, leading to single- and double-strand breaks [45]. Additionally, ionizing radiation liberates electrons from molecules, producing ions that can be chemically reactive [46], such as free radicals and reactive oxygen species, which can damage DNA. IR is ubiquitous in the environment. IR comes from naturally occurring radioactive materials and cosmic rays. IR also comes from common man-made sources of artificially produced radioisotopes, X-ray tubes, and particle accelerators. Because of its damaging effect on DNA, IR has long been linked to aging and certain age-related diseases [47]. IR induces oxidative stress, chromosomal damage, apoptosis, stem cell exhaustion, and inflammation, all of which are processes implicated in biological aging [47]. Consistent with our finding of IR-induced cellular senescence in IVDs, IR-exposed articular chondrocytes have also been shown to undergo senescence [48].

Our findings support AF cellular senescence as a contributor to IDD induced by IR. A previous study by Vamvakas et al. focused on the effects of IR on NP tissue. In a human NP cell culture model, they found that IR exposure or over-expression of p16 using lentiviral plasmid transfection resulted in a similar gene expression profile of replicative senescent cells derived by serial subculturing [49]. Upregulation was observed in matrix catabolic and inflammatory genes in IR-treated human NP cells, including MMP-1, -2, -3, -9, IL-6, IL-8, and IFN-c. Similar to this study, we found that IR-treated AF cell cultures upregulated MMP-1 and MMP-3 gene expression. We also found increased MMP-mediated disc aggrecanolysis in both the in vitro AF cell culture and the in vivo mouse model. IR-induced senescent AF cells in culture exhibited an overall decrease in total GAG content, consistent with the reduced GAG production in H202-induced senescent NP cells in culture reported previously [50]. However, it is interesting to note that IR-induced senescent AF cells in culture exhibited upregulation of expression of the key matrix genes, including collagen-1, collagen-2, and aggrecan. The increase in aggrecan mRNA expression and decrease in total GAG reflects either a disconnect between transcription and translation of aggrecan in AF cells or a loss of aggrecan protein due to increased degradation. Another possibility is that the aggrecan transcription is more sensitive to IR treatment than its protein translation step. Globally, our findings suggest an integrated mechanistic pathway between age-related and IR-induced IDD.

This study provides direct evidence of IR-induced genotoxic stress as a causative factor that accelerates IDD. Moreover, it reveals important clinical implications of the potential role of IR in the development of IDD in patients. Clinical considerations include the use of IR in the management of spinal tumors, imaging studies including CT scans, and CT-guided interventions such as injection, fluoroscopy (fluoroscopic-guided interventions such as kyphoplasty), and occupational exposure [51,52,53]. Beyond known links to the development of tumors, these patients may be further susceptible to disc matrix breakdown and IDD pathologies after IR exposure [33,41,42,51].

### Strengths and Limitations

Strengths of this study include the use of both in vitro and in vivo models, the corroboration of findings across species, and the establishment of IR as a simple tool to induce disc cellular senescence for future research. A limitation of this study is the use of in vitro cell cultures derived from rats and humans, while the in vivo model used was mice. Mice are too small to provide sufficient disc cells for our cell culture work. Another limitation is the use of whole-body IR exposure to a single IR dosage for our in vivo mouse model, which precludes us from clearly attributing the detrimental effects on disc tissue to the direct action of IR on disc tissue or to the indirect action of IR on other body tissues.

## 4. Materials and Methods

### 4.1. Protocol

This study was performed with University of Pittsburgh Institutional Animal Care and Use Committee (IACUC) approval (IACUC protocol 16067657, “The role of cellular senescence in intervertebral disc aging”) and University of Pittsburgh Institutional Review Board (IRB) approval (IRB protocol CR19070209-004, “Isolation and culture of human IVD cells and related tissues for examination of biological mechanisms of disc degeneration and treatment strategies”).

### 4.2. Rat AF Tissue Sample Collection and Cellular Isolation

Eight *F344* strain rats aged 1 year were housed 2 per cage in our animal facility with standard food and water in compliance with IACUC protocols. The rats were sacrificed, and AF tissue was surgically dissected from the spine. Tissue processing followed previously described protocols [13]. AF tissue was digested in 0.2% pronase (EMD Chemicals 53702, Darmstadt, Germany) for 60 min, followed by overnight digestion in 0.02% collagenase P (Roche Applied Science 11213872001, Basel, Switzerland) to obtain isolated AF cells. These primary rat AF cells were cultured in hypoxic conditions (37 °C, 5% CO_2_, 5% O_2_, and in buffered F12 medium supplemented with 10% FBS). Cells reached 65–85% confluence before stratification into control and three different radiation treatment groups. Cultured AF tissue samples were exposed to 0 Gy, 5 Gy, 10 Gy, or 15 Gy of IR via a Cesium 137 J.L device. Tissue incubation was subsequently continued under previous hypoxic conditions.

### 4.3. Human AF Tissue Sample Collection and Cellular Isolation

Human AF cells were isolated from the discarded cervical IVD tissue of 3 patients (42 ± 16 years, and average Thompson disc degenerative grade 2–3) undergoing elective surgical procedures for degenerative cervical disc disease. AF tissue specimens from individual patients were collected and pooled if more than one level was resected. Tissue processing followed previously described protocols [54]. AF was diced into small pieces and sequentially digested with 0.2% Pronase (EMD Chemicals 53702) in Ham’s F-12 medium supplemented with 10% FBS and 1% P/S for 1 hour, followed by 0.02% Collagenase- P (Roche Applied Science 11213872001) in Ham’s F-12 medium supplemented with 10% FBS and 1% P/S overnight in an incubator at 37 °C and 5% CO_2_. The isolated hAF cells were then seeded in T-75 flasks at a density of 600,000 cells/flask in 5 mL of resuspension media and cultured at 37 °C, 21% O_2_ and 5% CO_2_ in a humidified incubator. Cells were cultured to 65–85% confluence then stratified into control or 15 Gy radiation treatment groups using the same Cesium 137 J.L. device as the rat AF cells. Tissue incubation was continued under hypoxic conditions.

The NP was dissected from patient disc surgical specimens (19–59 years, mean = 42.2 years, and average Thompson degeneration grade 2–3).

### 4.4. CCK8 Cell Counting Assay

A CCK8 cell counting assay was used to measure cellular viability and proliferation rates 1, 2, and 5 days post–IR treatment. The CCK8 assay manufacturer’s instructions were followed, with 10 μL of CCK8 solution added per 100 μL of rat AF cell suspension. Using an optical density microplate reader, absorption at 450 nm wavelength was measured in all IR treatment and control group AF cell cultures.

### 4.5. Senescence-Associated β-Galactosidase Assay

A senescence-associated β-galactosidase (SAβ-gal) assay was performed using a Senescence Cells Histochemical Staining Kit (CS0030, Sigma, St. Louis, MO, USA) according to manufacturer’s instructions in a manner that has been previously described [46]. Briefly, rat AF cells were fixed for 7 min in fixation buffer at room temperature. Then the cells were washed twice with 2 mL of 1× PBS and stained with staining mixture overnight at 37 °C without CO_2_ (until the cells were stained blue). Finally, the cells were observed and imaged using brightfield microscopy at 40× *g* magnification (Nikon, Eclipse, E800, Melville, NY, USA). The percentage of β-galactosidase-positive cells was calculated by counting blue-stained cells against the total number of cells.

### 4.6. Transcriptional qRT-PCR Analysis of Senescence, Catabolic, and Anabolic Genes

Total RNA was extracted from cells using the Qiagen RNeasy Plus Micro Kit (Qiagen 74034). Next, qRT-PCR was performed to measure relative gene expression of senescent, anabolic, and catabolic genes of interest (Table 1) using iTaq™ Universal SYBR^®^ Green One-Step kit (Cat. No. 1725151; Bio-Rad, Hercules, CA, USA) and Bio-Rad iCycler IQ5 Detection System, utilizing a protocol previously described [54]. The threshold cycle number was determined by Bio-Rad software, and reactions were performed in duplicate. Finally, the relative mRNA expression levels were calculated using the comparative ∆∆Ct method, with Ct values normalized to mRNA levels of the *GAPDH* gene.

### 4.7. DMMB Assay for GAG Content

A colorimetric DMMB assay was performed to quantify total GAG matrix content in rat AF cell cultures in a manner previously described [35]. The GAG assay was performed 5 days after IR treatment to allow quantifiable GAG production. Extracellular matrix digestion buffer was added to cell culture plate wells and incubated at 37 °C (non-CO_2_ incubator) for 6 h before sample collection and DMMB assay. A standard curve created from known serial dilution concentrations of chondroitin sulfate was used to quantify GAG content, then normalized to the quantity of DNA per sample (Picogreen assay, Life Technologies P7589, Carlsbad, CA, USA).

### 4.8. Immunohistochemistry and IF Staining

IF to qualitatively analyze senescent, catabolic, and anabolic markers of interest in mouse disc tissues was conducted following a previously described protocol [55]. IVD tissue was isolated and fixed overnight at 4 °C in 2% paraformaldehyde. For IF staining, the tissues were cryoprotected with 30% sucrose in PBS overnight at 4 °C and then embedded in OCT(Tissue-Tek). Serial axial plane cryosections were cut at thicknesses of 5 µm. The tissue sections were rehydrated in PBS, permeabilized, and blocked with 0.25% Triton X-100, 10% goat serum, and 1% BSA in PBS for 30 min at room temperature. Following blocking, incubation with a primary antibody of interest was carried out overnight at 4 °C. The following primary rabbit antibodies of interest were used for IF staining: anti-P16 antibody (20 µg/mL, Cat. No. ab189034; Abcam, Cambridge, UK), anti-P21 antibody (1:250 dilution, Cat. No. ab188224; Abcam), anti-MMP-1 antibody (1:100 dilution, Cat. No. ab28196; Abcam), anti-MMP-3 antibody (1:200 dilution, Cat. No. ab39012; Abcam), anti-Collagen-1 antibody (1:200 dilution, Cat. No. ab34710; Abcam), anti-Collagen-2 antibody (1:200 dilution, Cat. No. ab34712; Abcam), anti-Aggrecan antibody (1:200 dilution, Cat. No. ab1031; Millipore, Burlington, MA, USA), and anti-HMGB1 antibody (1:200 dilution, Cat. No. ab18256; Abcam). The sections were then incubated with secondary antibodies (Cy3-conjugated goat anti-rabbit IgG, Cat. No. 111-165-003, Jackson ImmunoResearch Laboratories, Chester County, PA, USA) for 60 min at room temperature, according to the manufacturer’s protocols. Immunostained sections were imaged and analyzed using a Nikon instrument A1 confocal laser microscope and NIS-Elements microscope imaging software for antibody-specific staining in disc tissues.

### 4.9. Murine IR Model and Mouse Disc Tissue Extraction

Ten *C57Bl6* mice aged 5 months were housed in our animal facility. Using a previously described protocol, the mice were exposed to total body IR [56]. IR was delivered by a Cesium 137 J.L. device after mice were placed in an apparatus on a circular plexiglass pie plate without restrictions to mobility. Five mice were spun continuously on a turntable for one rotation total lasting 8 s while 3 Gy of IR was administered to ensure an even dosage to all animals. IR treatment was repeated once daily for a total duration of 1 week. Mice were examined daily by a dedicated laboratory technician for signs of illness. No illness was found before the mice were sacrificed 1 year later, and lumbar IVD tissue was isolated for analyses.

### 4.10. Western Blot Assay for Aggrecanolysis

Western blot was used to quantify protein extraction from mouse spines for aggrecan fragmentation using a previously described procedure [35]. Mouse spines receiving either 0 or 3 Gy of radiation were dissected for total lumbar disc tissue. Disc tissue was then incubated in 4 M guanidine HCl solution for 3 days followed by overnight digestion in chondroitinase ABC enzyme (10U ChABC/mL H_2_O, Sigma C3667-5UN). Disc protein extracts were then used for western blot for detection of aggrecan fragmentation, using Tris-HEPES 4–20% gradient gel (Thermo Scientific 25204, Waltham, MA, USA), Tris-HEPES-SDS Running Buffer (Thermo 28398), Tris-Glycine Transfer Buffer with 10% Methanol (Thermo Scientific 28380, Fisher Scientific A452-4), and TBST (Sigma-Aldrich T9039). Immunoblot detection was performed using 1:1000 rabbit polyclonal anti-Aggrecan primary antibody (Abcam ab36861), followed by 1:10000 anti-rabbit goat secondary antibody with HRP (Thermo Scientific PI-31460), and then chemiluminescent imaging (Thermo Scientific 34096 and Bio-Rad ChemiDoc MP). Immunoblot quantification was performed with densitometry analysis and local background subtraction. We used the anti-G1 antibodies (Abcam ab36861) to detect aggrecan fragments generated by MMP and ADAMTS proteolytic cleavage within the interglobular domain (IGD) of aggrecan as previously established [37].

### 4.11. Statistical Analysis

Statistical analysis was conducted using Prism statistical and graphing software. In addition, the two-tailed Student’s T-test was performed to analyze statistical differences and significance between IR-treated and control groups for various biochemical tests and assays.

## 5. Conclusions

Our study demonstrates that IR-induced senescent disc cells exhibit matrix homeostatic imbalance. This further confirms the role of DNA damage in driving IDD through cellular senescence. Our findings also have important clinical implications as ionizing radiation has potential adverse effects on spinal tissue, and hence patients undergoing IR-related treatment may be at increased risk of IDD and its sequelae.

## Figures and Tables

**Figure 1 ijms-23-04014-f001:**
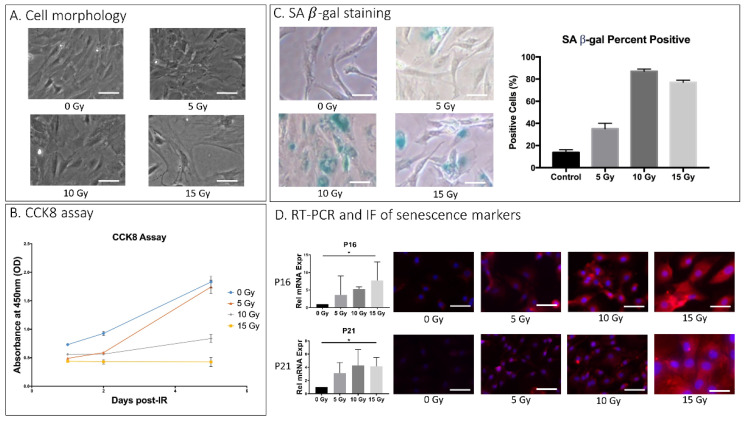
IR-induced AF cellular senescence. Rat AF cell cultures were exposed to a single dose of 0, 5, 10, and 15 Gy of IR and further incubated for 5 days before being analyzed for SASP. (**A**) Effects of IR on gross cell morphology as assessed by light microscopy. (**B**) Effects of dose and duration of IR treatment on cell proliferation in rat AF cell culture as quantitatively measured by CCK8 assay. Y-axis, proliferation level by absorbance. X-axis, time in days post–ionizing radiation. (**C**) Senescence-associated β-galactosidase (SA-βgal) assay of rAF cell cultures 5 days post-treatment with different IR doses. Cultures treated with 10 or 15 Gy IR resulted in 80% of cells stained positive for SA-βgal. (**D**) IR increased expression of cellular senescence markers, p16 and p21 in a dose-dependent manner as measured by qRT-PCR for mRNA levels (left) and by immunofluorescence (IF) for protein levels (right). Data shown as an average of *n* = 3 with one standard deviation. On microscopic images, the white bar for scale measures 20 μm. * *p* ≤ 0.05.

**Figure 2 ijms-23-04014-f002:**
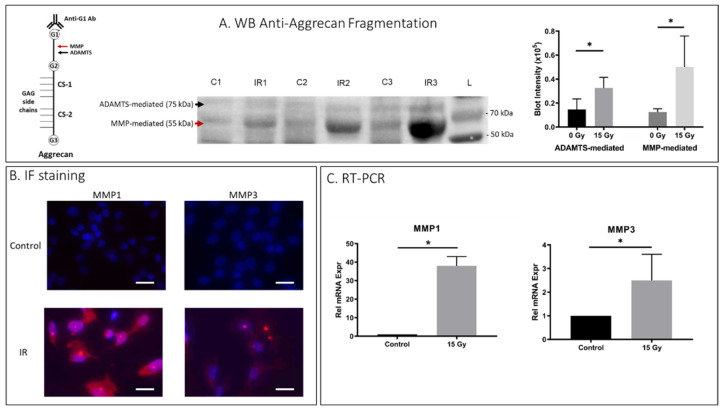
IR treatment increased matrix breakdown in rat AF cells. Rat AF cell cultures were treated with 15 Gy IR and incubated for 5 days to establish senescence. (**A**) Western blot (WB) analysis of anti-aggrecan fragmentation from 3 control and 3 IR AF samples showed elevated anti-aggrecan fragmentation mediated by ADAMTS (75 kDA) and MMP (55 kDa) in IR-treated AF cell samples. “C” designates controls, “IR” designates ionizing radiation treated, and “L” designates protein ladder. Left side, a schematic representation of aggregate consisting of the core aggrecan protein bound to GAG side chains at the chondroitin sites (CS-1, CS-2), and the MMP- and ADAMTS-mediated cleavage site within the interglobular domain residing between the G1 and G2 domain of aggrecan are indicated with arrows. Right side, quantification of western blot band intensity is depicted in graphs on right. (**B**) IF staining demonstrated elevated MMP1 and MMP3 protein expression (red) in IR-treated AF cells compared to untreated cells. (**C**) qRT-PCR also confirmed elevated MMP-1 and -3 gene expression in IR-treated AF cells compared to untreated cells. Data shown as an average of n = 3 with one standard deviation. On microscopic images, the white bar for scale measures 20 μm. * *p* ≤ 0.05.

**Figure 3 ijms-23-04014-f003:**
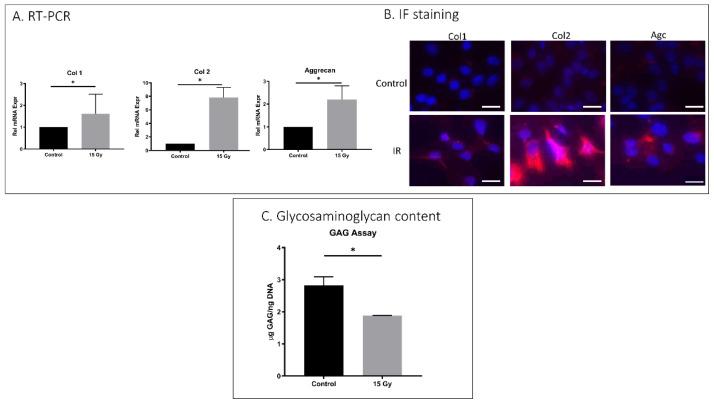
IR treatment effects on matrix anabolism in rat AF cells. Rat AF cell cultures were treated with 15 Gy IR and incubated for 5 days to establish senescence. (**A**) qRT-PCR revealed increased mRNA expression of collagen-1, collagen-2, and aggrecan in IR-treated AF cells compared to untreated AF cells gene expression. (**B**) Immunofluorescence showed increased collagen-2 but unchanged collagen-1 and aggrecan protein expression with 15 Gy of ionizing radiation administration. (**C**) DMMB assay for total glycosaminoglycan (GAG) content showed a decrease in GAG in IR-treated AF cells compared to untreated control. Data shown as an average of n = 3 with one standard deviation. * *p* ≤ 0.05.

**Figure 4 ijms-23-04014-f004:**
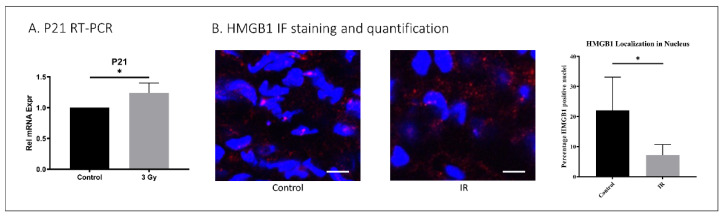
IR whole-body exposure induces disc cellular senescence in mice. (**A**) Whole-disc tissue p21 gene expression was increased in IR-treated mice compared to untreated control. (**B**) HMGB1 immunofluorescence staining (red) and quantification of nucleus (blue, DAPI) localization revealed decreased nuclear expression of HMGB1 in disc tissue of IR-treated compared to untreated mice. Data shown as an average of n = 5 with one standard deviation. On microscopic images, the white bar for scale measures 20 μm. * *p* ≤ 0.05.

**Figure 5 ijms-23-04014-f005:**
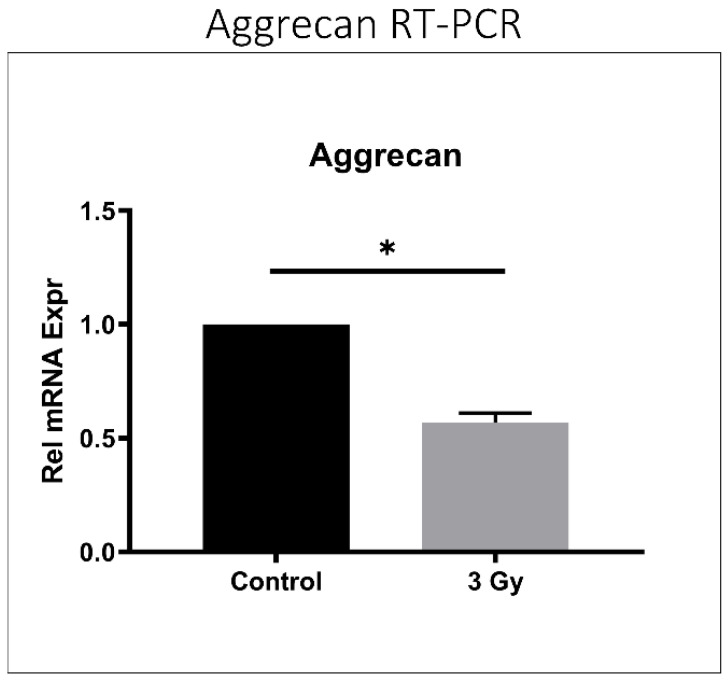
IR treatment reduced aggrecan gene expression in mouse intervertebral discs. RT-PCR quantification of murine aggrecan relative gene expression after the administration of 3 Gy ionizing radiation is depicted compared to control. Data shown as an average of n = 5 with one standard deviation. * *p* ≤ 0.05.

**Figure 6 ijms-23-04014-f006:**
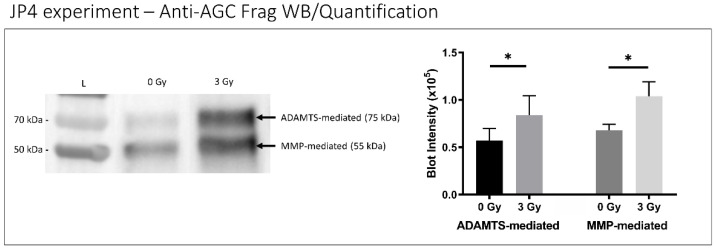
IR treatment increased disc aggrecanolysis in mice. A representative western blot of whole- disc tissue extract showing aggrecan fragments generated from MMP-mediated (55 kDa band) and ADAMTS-mediated (75 kDa band) activities are shown (left) with quantitative band intensity results (right) exhibited. Western blot “L” designates protein ladder with “0 Gy” and “3 Gy” representing radiation treatment dose groups. Data shown as an average of n = 5 with one standard deviation. * *p* ≤ 0.05.

**Table 1 ijms-23-04014-t001:** qRT-PCR primers used for quantifying genes involved in senescence and matrix anabolism and catabolism. Forward (FW) and reverse (RV) sequences for human PCR primer genes used in qRT-PCR assays for matrix proteins, matrix catabolic proteins, and cellular senescence markers. A, T, C, and G letters in sequences represent base pairs.

Gene (Human)	FW Sequence (5′ to 3′)	RV Sequence (3′ to 5′)
Aggrecan	ATACCCCATCCACACGCCCCG	GCGAAGCAGTACACATCATAGG
Col 1	GCCAAGAAGACATCCCTGAAG	TGTGGCAGATACAGATCAAGC
Col 2	GTGGAGCAGCAAGAGCAAGGA	CTTGCCCCACTTACCAGTGTG
MMP1	TCTTTATGGTCCAGGCGATGAA	CCTCTTCTATGAGGCGGGGAT
MMP3	GGTACAGAGCTGTGGGAAGTC	GATGAGCACACAACCACACAC
P16	AATCTCCGCGAGGAAAGC	GTCTGCAGCGGACTCCAT
P21	TCCACAGCGATATCCAGAC	GGACATCACCAGGATTGGA

## Data Availability

The data presented in this study are openly available upon request.

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
