# Peer review of "Ionizing Radiation Induces Disc Annulus Fibrosus Senescence and Matrix Catabolism via MMP-Mediated Pathways"

_ijms, 2022, doi:10.3390/ijms23074014_

Round 1

Reviewer 1 Report

The last sentence of the abstract mentions "chemotherapy."  Did you mean to say "ionizing radiation" instead?  Chemotherapy is not mentioned anywhere else in the article except in Reference 24. 

The sixth sentence of the introduction states that loss of disc PG is a result of matrix catabolic activities, which to me sounds like a purely chemical process, but in reality isn't loss of disc PG also as a result of mechanical damage? If so, one could insert the words "mechanical damage and" before "increased matrix catabolic activities.."

In the first paragraph of Results, should "CCK8" be spelled out first before using the abbreviation?

Line 89 - the word "increased" should be changed to "increase". 

Reviewer 2 Report

Overall, this study provides novel basic understanding of the role of proteoglycan catabolism in IR-induced IDD. In general, I recommend that the authors provide more clear microscopy images with a appropriate annotations before this manuscript is acceptable for publication. Similarly, multiple anti-aggrecan antibodies should be considered to better suggest ADAMTS- vs MMP- mediated degradation. Lastly, the authors do not provide sufficient introduction or discussion regarding the role of senescence on fibrosis/matrix degradation.

Introduction:

  • Line 33-34: Proteoglycan is not specific enough – which proteoglycans are seen in IDD and are known to degenerate? In what way?
  • Line 39-40: “DNA damage…. Primary cause of aging” – while a major contributor, this is an overstatement. Please revise.
  • Please give a brief review on what is known regarding matrix (specifically proteoglycan) degradation and DNA damage/senescence, even if it is outside the field of IDD.

Results:

  • Figure 1: Please include scale bars on all microscopy images. Similarly, zooms shown inefficiently capture changes in cell morphology as claimed.
  • 1D: Microscopy images unclear – please shown higher magnification
  • Figure 2: It is unclear based on your annotations that the western blot is anti-aggrecan. Please specify. Similarly, please provide citations for ADAMTS- and MMP-mediated fragmentation of aggrecan causing two distinct MW-sized fragments. Please discuss this in the results section briefly before introducing Figure 2.
  • Justification of antibodies used and specific epitopes should be included in the methods section, minimally,
  • Figure 2: Please perform an anti-ADAMTS western and qRT-PCR, similar to MMP studies.
  • Figure 2B: microscopy images are out of focus and unclear – similarly, scale bar is too small. Please revise.
  • Complementary experiment: Please perform gelatin-zymography to get a clearer picture of MMP-influence in IR treatment. The authors do not justify their choice in MMP1 and MMP3 for qPCR and microscopy-based evaluations.
  • Figure 4: Please include scale bars
  • Figure 4: Microscopy images are low quality and inefficient magnification to show nuclear vs. cytoplasmic localization.

Round 2

Reviewer 2 Report

The authors have adequately addressed all concerns.